# Equinoctial Asymmetry in Solar Quiet Fields along the 120° E Meridian Chain

Yingyan Wu [1,*], Libo Liu [2,3,4] and Zhipeng Ren [2,4,5]

1   Institute of Earthquake Forecasting, China Earthquake Administration, Beijing 100036, China
2   Key Laboratory of Earth and Planetary Physics, Institute of Geology and Geophysics, Chinese Academy of Sciences, Beijing 100029, China; liul@mail.iggcas.ac.cn (L.L.); zpren@mail.iggcas.ac.cn (Z.R.)
3   Heilongjiang Mohe Geophysical Observatory, Institute of Geology and Geophysics, Chinese Academy of Sciences, Beijing 100029, China
4   College of Earth and Planetary Sciences, University of the Chinese Academy of Sciences, Beijing 100049, China
5   Beijing National Observatory of Space Environment, Institute of Geology and Geophysics, Chinese Academy of Sciences, Beijing 100029, China
*   Correspondence: wuyyan79@126.com

**Abstract:** Equinoctial asymmetry of the range of the solar quiet day variation (Sq) of the horizontal geomagnetic field (H) has been found in some low latitude geomagnetic observatories. This study conducted an investigation of its latitude distribution and the relationship with the solar cycle by using the H field measurements from six observatories along the 120° E meridian chain in the years 1957–2013. Results illustrate a significant equinoctial asymmetry of the SqH range at all observatories. Three main features were identified. First, the signature of the equinoctial asymmetry of the SqH range is opposite for observatories located at the northern and southern sides of the Northern Hemisphere Sq current focus. It shows larger values around spring than autumn equinox at southern observatories, and the converse is seen at northern observatories. Second, the asymmetry increases with the distance from the Sq current focus, suggesting the stronger sensitivity of the distant observatories than observatories around the focus. The result of linear fitting presents a positive dependence of the asymmetry coefficient on geographic latitude, with a reversal of the asymmetry occurring at 28.1° N near the focus of the average Sq current. Third, there is no obvious dependence of the equinoctial asymmetry of the SqH range on solar activity, suggesting a possible cause from some regional factors related to the ionospheric dynamo process.

**Keywords:** solar quiet variation; equinoctial asymmetry; latitudinal distribution



## 1. Introduction

During geomagnetic quiet days, a regular diurnal variation—with an amplitude of a few tens of nanotesla (nT)—of the Earth's geomagnetic field, known as the solar quiet day variation (Sq), is observed at middle and low latitudes. It is primarily originated from two current whorls driven by atmospheric tidal winds and controlled by the ionospheric dynamo currents, flowing in the E region of the ionosphere at altitudes of 90–150 km: counterclockwise in the northern hemisphere and clockwise in the southern hemisphere [1–4]. Its daily range (or amplitude), phase (or shape), and the foci (or centers) of the Sq's ionospheric equivalent current exhibit variations on multiple timescales—diurnal, seasonal, and over a solar cycle [5–8].

Seasonal variations of Sq have been extensively studied in the past decades. The Sq field amplitude shows maxima during equinoctial months at low and equatorial latitudes [1,9–11]. Similar equinoctial peaks of the Sq current intensity were also identified in simulation results, which were possibly caused by tidal waves in the lower atmosphere [12–15]. Moreover, the position of the Sq current focus also showed seasonal shifts in latitude and local time (LT) during equinoctial months [16–24].

Prior investigations generally combined spring and autumn months to represent the "equinox season", as defined by Lloyd (1874) [25]. However, a few studies differentiated between spring and autumn months. Howe (1950) [26] found that the amplitude of the horizontal (H) component of Sq (SqH) was smaller in September than in March at the Honolulu observatory (HON, 21.3° N, 201.9° E). Wulf (1963, 1965) [27,28] compared seasonal variations of the SqH range between the HON, San Juan (SJG, 18.4° N, 293.9° E), and Tucson (TUC, 32.2° N, 249.3° E) observatories and found a larger maximum in March than in September at HON and SJG, but a reversed situation at TUC. He suggested that it was caused by an anomalous seasonal variation of the large-scale lower-ionospheric circulation. Analyzing data from eighteen observatories located at low-to-high latitudes, Chulliat et al. (2005) [29] confirmed the significant SqH equinoctial asymmetry at HON and TUC. They suggested that this asymmetry was only weakly correlated with solar activity but was more likely the result of the ionospheric dynamo induced by seasonal asymmetry of lower-thermospheric winds. Recently, similar SqH equinoctial asymmetry was reported by Falayi (2014) [30] at an equatorial observatory, Addis Ababa (AAE, 9.0° N, 38.8° E). Furthermore, equinoctial asymmetry of the Sq equivalent current also has been found by the method of spherical harmonic analysis. Takeda (2002) [31] observed different phases of the Sq equivalent current in March and September and attributed this to the equinoctial asymmetry of tidal winds in the upper atmosphere. Yamazaki et al. (2010) [14] examined the month-to-month variation of the Sq current intensity in East Asia and discovered a spring–autumn asymmetry, with a larger intensity in spring.

Obviously, based on few stations in previous ground observations, the understanding of the detailed features of the SqH equinoctial asymmetry is limited in former studies. Its latitudinal distribution and its relationship with the current center and solar activity need to be better clarified. Therefore, the purpose of this study was to investigate the latitudinal distribution of the equinoctial asymmetry of the range of SqH by using hourly observations of the horizontal component of the geomagnetic field, acquired at six geomagnetic observatories located along the 120° E meridian chain. Their proximity locations allow us to examine more detailed characteristics of the equinoctial asymmetry of the SqH at the Sq current region of the Northern Hemisphere, located in East Asia, than previous studies. Additionally, we investigated the relationship between the SqH and the solar activity using data from the years 1957–2013.

## 2. Data and Calculations

In this study, we used hourly measurements of the horizontal component of the geomagnetic field from six geomagnetic observatories located in the Northern Hemisphere along the 120° E meridian chain. Five are in China—Beijing Ming Tombs (BMT), Lanzhou (LZH), Chengdu (CDP), Wuhan (WHN), and Guangzhou (GZH)—and one—Muntinlupa (MUT)—is in the Philippines. The five Chinese observatories are member stations of the Chinese meridian project, more information on which can be found on the website: "https://data.meridianproject.ac.cn/about-us/" (accessed on 18 September 2021). The MUT station is part of the 210 magnetic meridian chain, listed in the catalogue of stations on "https://stdb2.isee.nagoya-u.ac.jp/mm210/station.html" (accessed on 18 September 2021). Here, the data of these stations are downloaded from the World Data Center for Geomagnetism (Edinburgh) on site: "ftp://ftp.nmh.ac.uk/" (accessed on 18 September 2021). As declared by WDC (Edinburgh), the geomagnetic data are collected from operating observatories and INTERMAGNET (International Real-time Magnetic Observatory Network), and keep the data policy of IAGA (International Association of Geomagnetism and Aeronomy) to publish filtered one-minute data and one-hour data. As known from the introduction of the observatories on WDC (Edinburgh) and INTERMAGNET, the instrumental system including vector fluxgate magnetometer and scalar magnetometer is equipped to obtain the absolute observed values of the magnetic field.

Theses six observatories are distributed over about 25° in geographic latitude (14–40° N) with an interval of 5–8°, except between CDP and WHN. Their coordinates

are listed in Table 1. Figure 1 shows their positions relative to the Northern Hemisphere Sq current system. This Sq current diagram describes an average ionospheric equivalent current, which is derived by the commonly used spherical harmonic analysis technology and by using the data of the yearly averaged solar quiet day variation of observatories in geographical longitude from 90 to 130 degrees in 2000—hence, with no focusing on its daily and monthly variation. CDP and WHN are near the focus of the Sq current. BMT and LZH will hereafter be referred to as "northern stations", while GZH and MUT will be referred to as "southern stations".

**Table 1.** Coordinates and time coverage of the geomagnetic observatories used in this study.

| Site | Geographic Coordinates | Geomagnetic Coordinates | Time Coverage |
|---|---|---|---|
| BMT | 40.3° N, 116.2° E | 29.9°, 186.8° | 1996–2013 |
| LZH | 36.1° N, 103.9° E | 25.7°, 175.9° | 1980, 1986, 1989–1992, 1995, 1997–2011 |
| CDP | 31.0° N, 103.7° E | 20.6°, 175.7° | 1995–2002, 2005–2007 |
| WHN | 30.5° N, 114.6° E | 20.1°, 185.62° | 1980, 1995–2002, 2005–2007 |
| GZH | 23.1° N, 113.3° E | 12.7°, 184.6° | 1960–1993, 2003–2009 |
| MUT | 14.4° N, 121.0° E | 4.2°, 192.2° | 1957–1959, 1963–1972 |

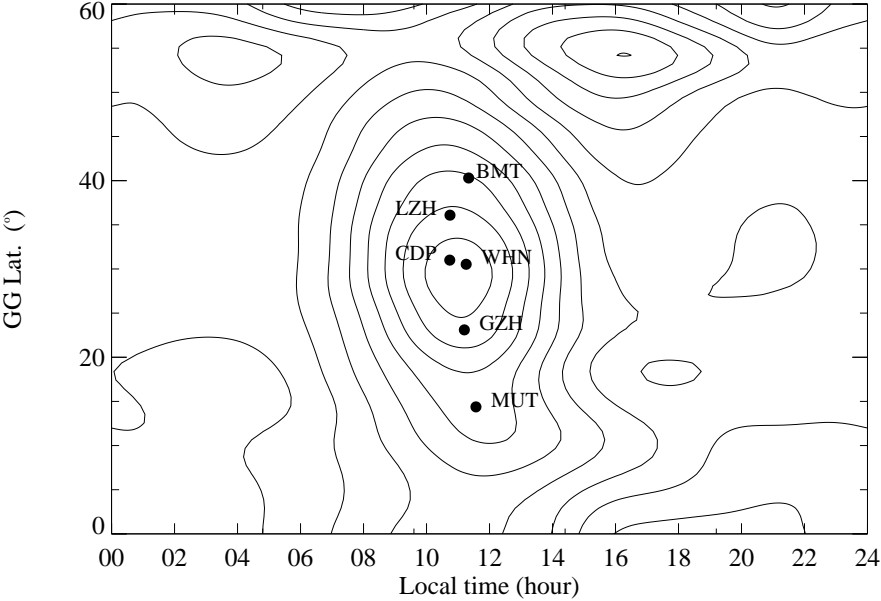

**Figure 1.** Schematic diagram of the geomagnetic observatories' locations used in this study, along the 120° E meridian chain, relative to the Northern Hemisphere solar quiet (Sq) current focus. "GG Lat". represents the geographic latitude.

Figure 2 shows the variation of solar radio flux at 10.7 cm (F107) in solar flux units (1 sfu = $10^{-22}$ W m$^{-2}$ Hz$^{-1}$) [32] and the yearly coverage of the geomagnetic field measurements used in this study. As we can see, the full dataset covers five solar cycles from 1957 to 2013. None of the observatories were operational during the complete period; however, time coverage is sufficient for investigating the latitudinal distribution of the SqH equinoctial asymmetry because of the good temporal overlap between the northern stations (1995–2007) and between both southern stations (1963–1972). These common periods are used to compare observatories on the same side of the Sq current focus.

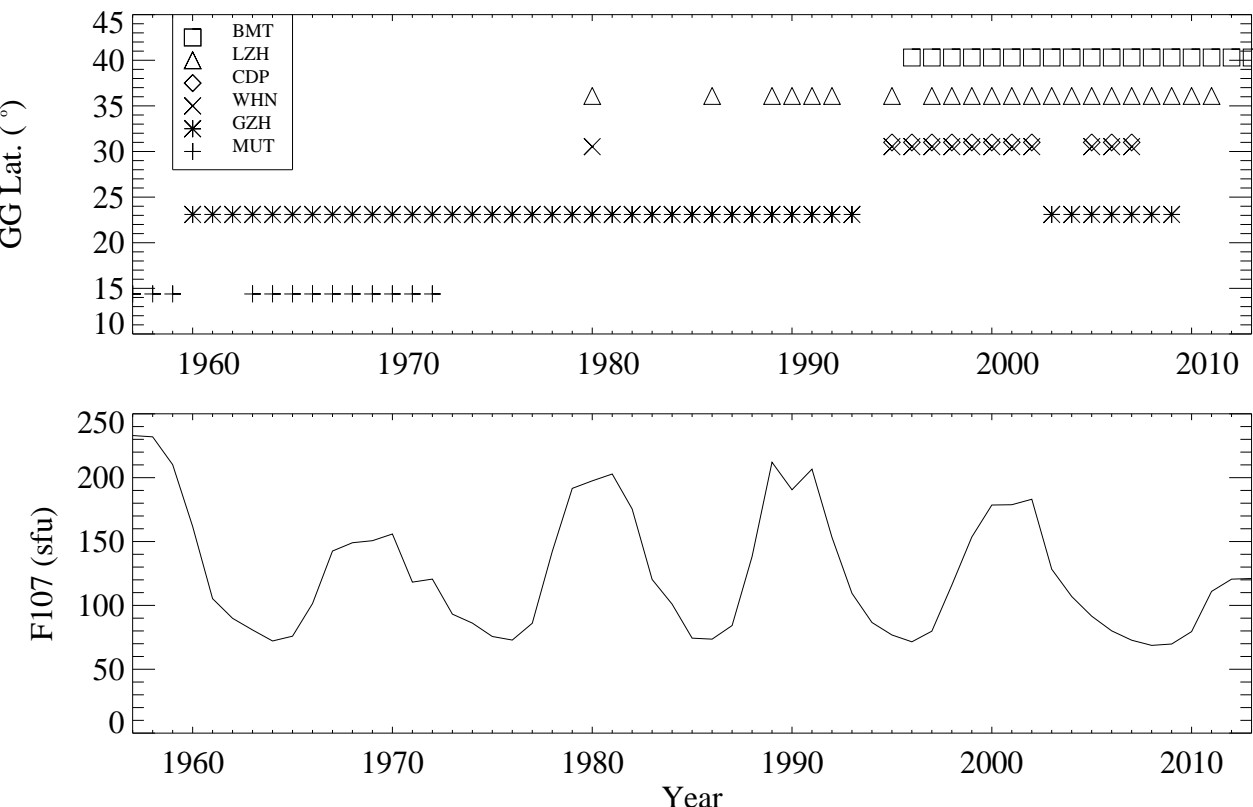

**Figure 2.** (**Top**): Yearly coverage of the observations used in this study at each observatory. (**Bottom**): Yearly averages of the 10.7 cm solar radio flux (F107) over the full time period (1957–2013), downloaded from the ftp server: ftp://ftp.gfz-potsdam.de/pub/home/obs/Kp_ap_Ap_SN_F107 (accessed on 18 September 2021).

To calculate SqH, hourly data are first accumulated to obtain daily values. The baseline value derived from the average "midnight" values between 23:00 and 02:00 LT and the secular changes calculated with a detrending method [33] are both removed from the daily measurements. Then, the SqH is calculated by averaging the results of the five international quiet days in each month [1], which is derived by GFZ Potsdam from the Kp index. From the daily SqH values, we further derive monthly and yearly averaged SqH values, shown in Figure 3. Marked diurnal and seasonal variations are clearly visible, and the standard deviations (d) as an error value for each hour are also plotted in red bars. In the right panel of Figure 3, the diurnal variations of the yearly averaged SqH of the southern stations (MUT and GZH) show a marked enhancement around 12:00 LT, with a magnitude decreasing closer to the Sq current focus. While, the strong enhancement as seen at noontime at the equatorial station MUT clearly reflects the contribution of equatorial electrojet (EEJ), which also could produce an impact on low latitude stations and could probably result in the misunderstanding of the characteristics of the Sq focus [11,34] However, curves of the diurnal variations are much more smooth at "central" stations (WHN and CDP). Conversely, the diurnal variation pattern of the yearly averaged SqH at northern stations LZH and BMT shows a strong decrease between 09:00 and 12:00 LT.

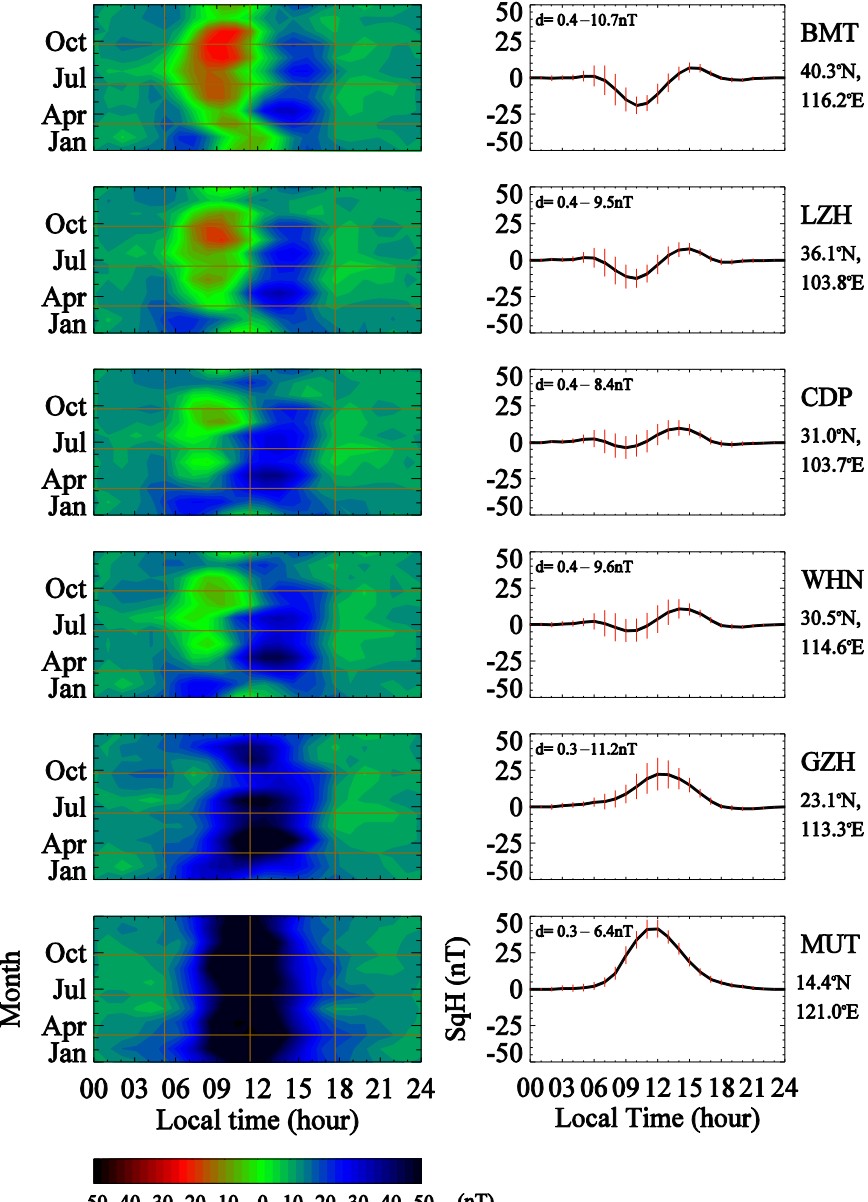

**Figure 3.** Diurnal variations of monthly (**left panel**), yearly (**right panel**) averaged SqH and associated standard deviations d (in red short lines) at each observatory, sorted by latitude from north (**top**) to south (**bottom**) of the Sq current focus.

In the left panel of Figure 3, average monthly SqH, derived from all available years in each station, are contoured, which shows the peak amplitude of the SqH in the equinoctial months. Additionally, the extreme values appear to be drifting in LT, occurring increasingly late from winter to summer at southern stations but early at northern stations. These combined features can fully correspond to the universal time (UT) variability of the Sq and to its ionospheric dynamo current [2,33,35]. Additionally, the SqH amplitude is clearly different between spring and autumn, presenting the asymmetry between the equinoxes.

Next, daily ranges of the monthly average of the SqH over 1957–2013 are calculated as the difference of the extremes: A = $H_{max}$ − $H_{min}$, plotted with black curves in Figure 4. They show significant month-to-month variation and a noticeable 11-year variation corresponding to the solar cycle, as shown by the 10.7 cm solar radio flux (F107 index) (Figure 4, top). It is similar to the variation of the intensity and focus of the Sq current and the correlation with solar activity [1,36–39].

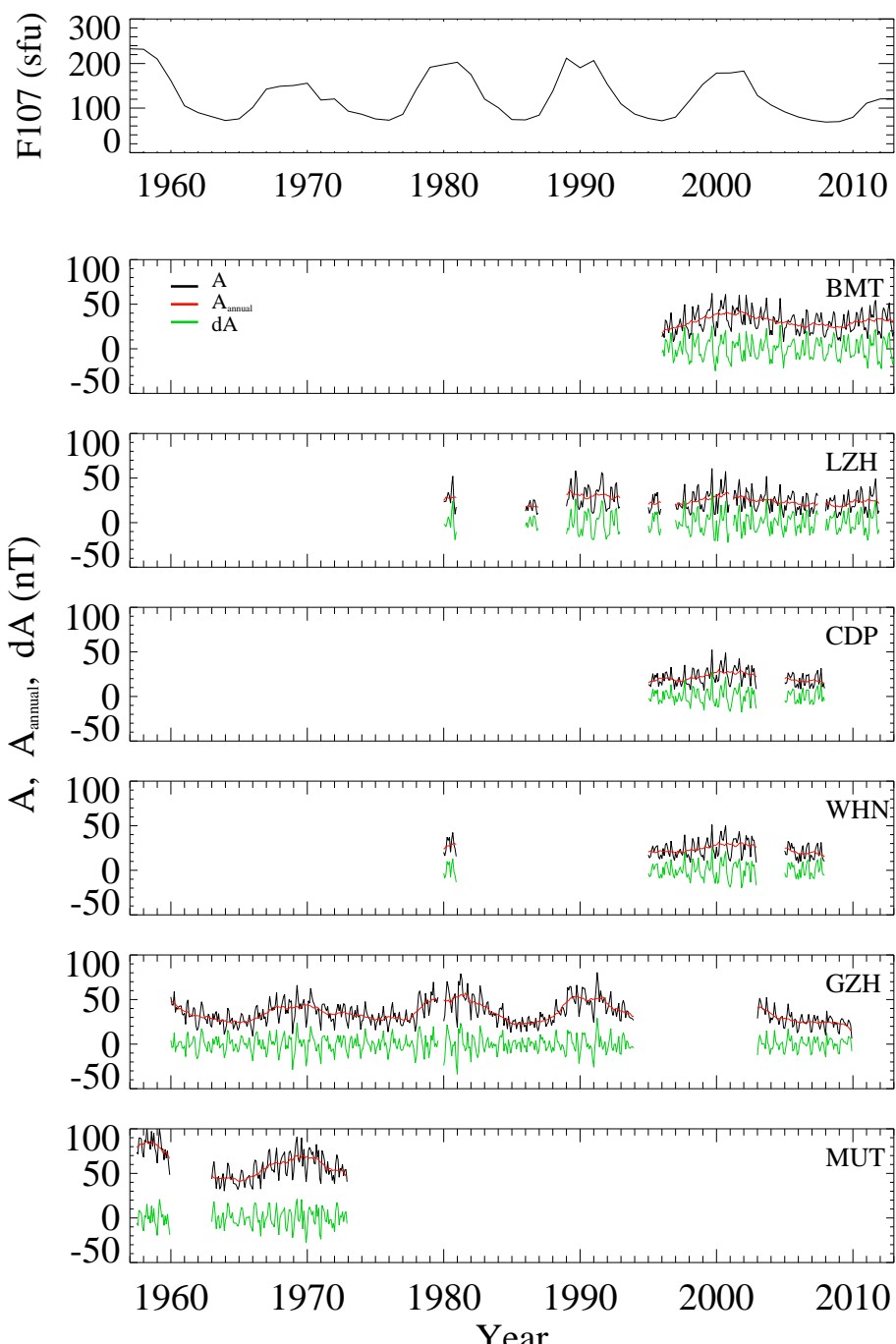

**Figure 4.** (**Top**): Yearly averaged F107 index (1957–2013). (**Bottom**): Month-to-month variations at each observatory of the SqH daily range (A, black curves); 12-month smoothing of the SqH daily range (A$_{annual}$, red curves) and their variability (dA, green curves).

Then, the annual value should be calculated to remove the contribution of the solar cycle on A. As is well known, the method of moving average is a common low-pass filter technique. The effect of the moving window size could not be neglected [11,40–42]. From their analysis, for a signal with period T, using 2T as the moving window was more effective in eliminating the periodic T signal than a 1T moving window. Therefore, we performed the analysis with both a 12-month and 24-month moving window, and we confirm that there is no essential difference between the results of these two windows. Thus, in this study, the 12-month moving window was chosen and derives the 12-month smoothed SqH range, noted with A$_{annual}$ and shown by red curves in Figure 4, in good agreement

with the F107 index. The month-to-month range variability is finally derived as a residual: dA = A − A$_{annual}$, and it is plotted in Figure 4 with green curves, which show successful removal of the solar activity contribution.

## 3. Results

The residual dA is then used to investigate seasonal variations of the SqH daily range. In the left panel of Figure 5, monthly dA values derived for each year are plotted for each observatory (superimposed black curves). While the spread of the black curves suggests only small year-to-year variations, marked seasonal variations occur every year. The dA value reaches peak around the equinoxes and is smallest in winter at all observatories, including CDP and WHN located near the Sq current focus. The right panel in Figure 5 shows the average dA variations, the associated standard deviations (SD) d, and the standard error of the mean (SEM, where SEM equals $SD/\sqrt{n}$, and n is the number of the stacked curves) [11,40] for each month. Seasonal variation—equinox maximum and winter minimum—is clearly visible at each observatory. Yearly dA ranges (R) are calculated as the difference of the extreme monthly values. We obtain R values of 27.4, 23.8, 22.3, 30.4, 30.0, and 29.4 nT for MUT, GZH, WHN, CDP, LZH, and BMT, respectively, indicating small differences between the observatories. Corresponding monthly standard deviations (*d*) are within 4.4–12.0, 4.7–8.0, 3.3–6.0, 3.4–7.3, 3.7–7.9, and 4.0–7.7, respectively. This indicates significant monthly variations of dA from year to year.

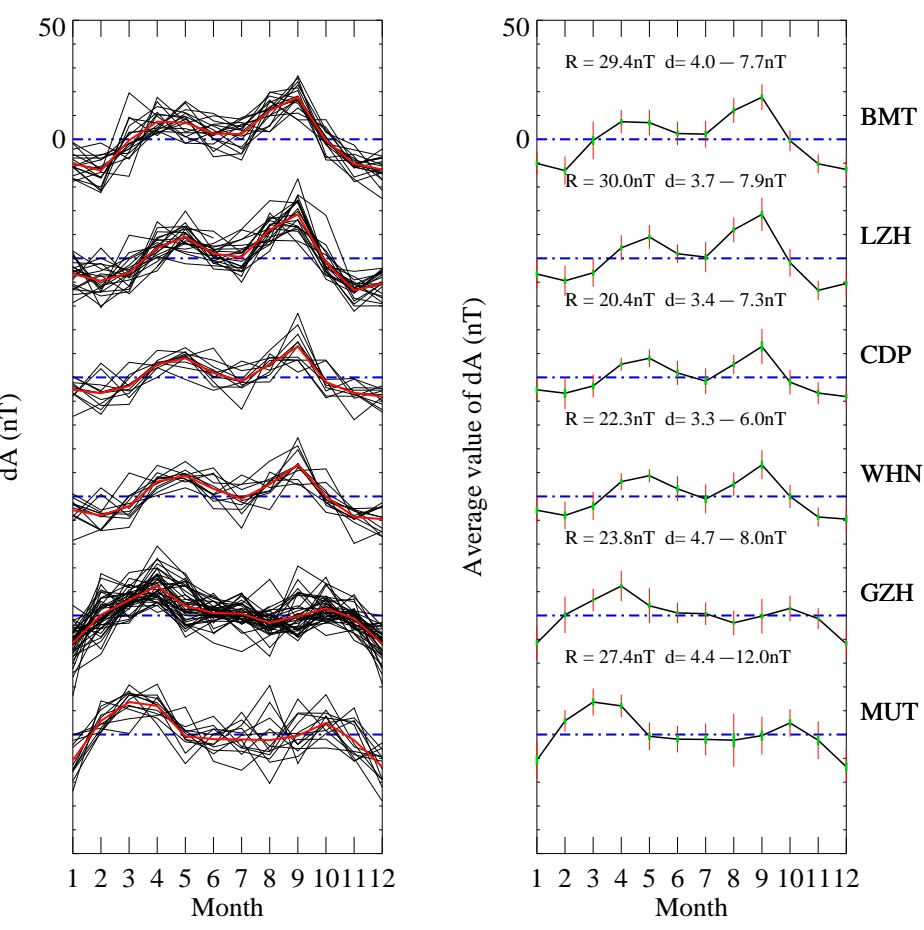

**Figure 5.** Seasonal variations of the SqH range (dA) over 1957 to 2013. Left panel: stacking curves of the monthly dA for all available years at each observatory. Right panel: average dA over all available observation years, associated standard deviations d (in red short lines), and SEM (in green bars); values of the yearly range R and of the standard deviation d are indicated.

Equinoctial asymmetry in each year is then defined as the difference of dA averages around the spring (March–April) and autumn (September–October) equinoxes: $AA = \overline{dA_{autumn} - dA_{spring}}$. To represent the equinoctial asymmetry variability, an "asymmetry coefficient" is further defined as $(dA_{autumn} - dA_{spring})/(dA_{autumn} + dA_{spring})$, and its values at each observatory are shown in Figure 6. By construction, the asymmetry coefficient absolute values cannot exceed $|\pm 1|$. A positive value of $AA$ or of the asymmetry coefficient indicates a greater SqH range in autumn than in spring (red bars), while negative values (blue bars) show reversed asymmetry, with smaller SqH range in autumn than in spring.

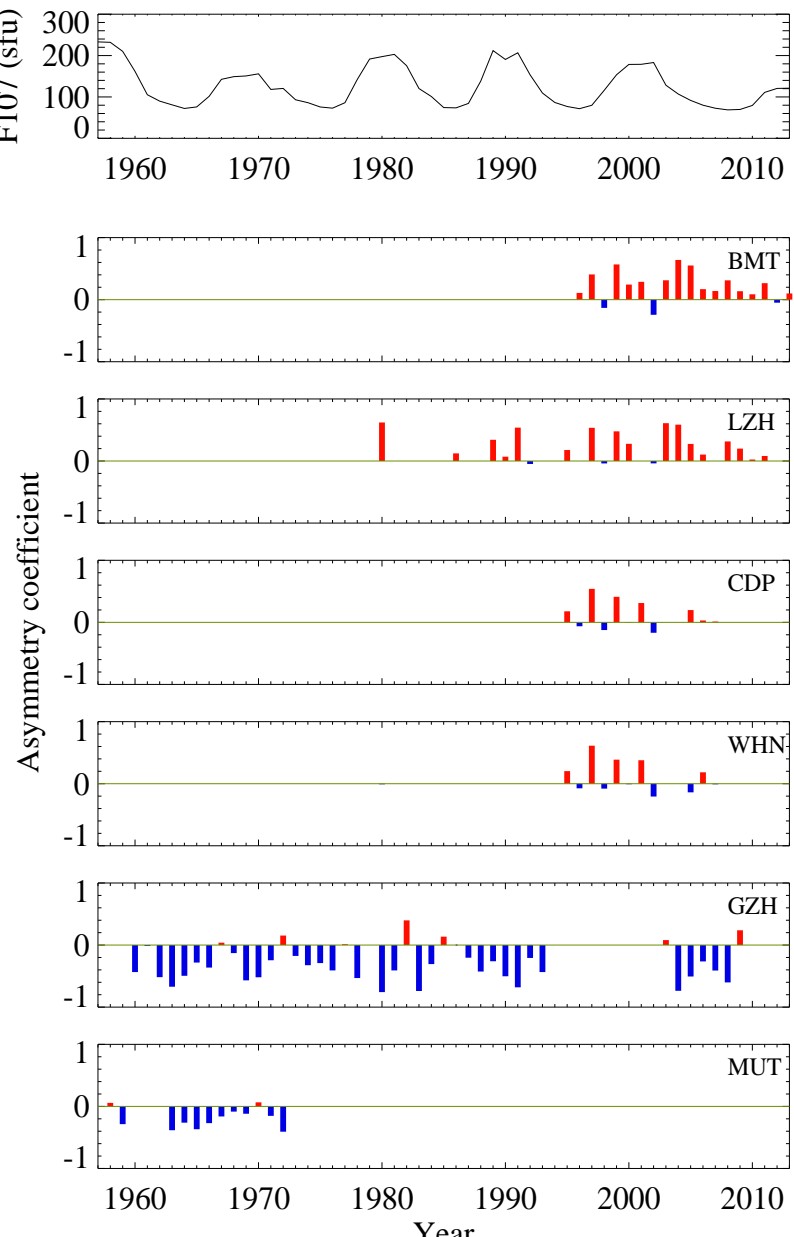

**Figure 6.** (**Top**): yearly averaged F107 index over the period from 1957 to 2013. (**Bottom**): Equinoctial asymmetry coefficient calculated at each observatory for years with available observations and shown in red and blue bars for the positive and negative values respectively.

The asymmetry coefficient is generally large—regardless of the year or the observatory—reaching, for example, −80% at GZH in 1980. The latitudinal dependence of the equinoctial asymmetry is clearly seen in Figure 6. Most notably, variations of the asymmetry coefficient exhibit the same behavior at observatories located on the same

side of the SqH current focus. Additionally, the asymmetry coefficient, generally positive at northern observatories (with a few exceptional years that displayed negative values, as shown in the blue box—for example, in years 1998, 2002, and 2012 at BMT station), changes sign around the Sq current focus (near WHN) and becomes mostly negative at southern observatories (also with a few exceptions that displayed positive values, as shown in the red box—for example, in years of 1958 and 1970 at MUT station). This means that dA is usually larger in spring than in autumn in most years at southern observatories but that the situation is reversed at northern observatories. Moreover, it is also noticed that the intensity of the asymmetry coefficients is very similar between both northern stations BMT and MZL; however, for the southern stations, it is obviously a smaller value in the equator station MUT than in the GZH station, probably related to the EEJ.

When comparing with solar activity, equinoctial asymmetry is quite stable at each observatory. Asymmetry values are mostly negative at southern observatories and positive at northern observatories. Obviously, they do not appear to depend on solar cycle, as increases or decreases in the asymmetry coefficient seem to occur during any phase of the solar cycle (Figure 6). For example, large asymmetry coefficient values are found during high solar activity years—negative in 1969 at GZH and positive in 1980 at LZH—or during low solar activity years, such as 1964 at GZH. Similarly, large values are found during both the descending—in 1983 at GZH or in 2005 at BMT and LZH—and ascending—in 1998 at BMT—phases of the solar cycle. In addition, a few near-zero asymmetry coefficient values are recorded at each observatory. About ten percent of cases from all measurements show values of the asymmetry coefficient that are less than 0.05. This implies that in these years the SqH range is comparable in spring and in autumn.

Figure 7 shows the variations (black dots) and SEM (red bars) of the equinoctial asymmetry *AA* and of the asymmetry coefficient—both averaged at each observatory over all available measurements—as a function of geographic latitude. The linear fit to the asymmetry coefficient is also shown (green line in the right panel), which has a median value of the errors of A and B (evaluated from the differences between the linear fit value and the measurement) of about $-1.14$ and $-7.54$, respectively. Equinoctial asymmetry is clearly visible along the meridian chain: the maximum value of the averaged AA is about $-4$ nT at "southern station" (GZH) and 5 nT at "northern station" (BMT), and the minimum value is about 1.5 nT presented at "central stations" (WHN and LZH). Accordingly, the maximum averaged asymmetry coefficient is about $\pm30\%$, and the minimum value is about 10%. Both quantities increase with geographic latitude and with increasing distance between the observatory and the Sq current focus, possibly suggesting that the distant observatories are more sensitive to the asymmetry mechanism than those close to the Sq current focus. Remarkably, a reversal of the equinoctial asymmetry sign is seen between the northern (positive) and southern (negative) observatories. Additionally, the latitude of the reversal is about 28.1°, which is located near the focus of the focus of the average Sq current (as seen in Figure 1) because of the linear fitting between the asymmetry coefficient and the geographic latitude. Notably, this reverse latitude is only a statistical average result from all the data. The linear fitting results will generally change if the SEM and the confidence interval are considered. After doing some tests with different confidence intervals, we found little change in coefficient A, suggesting the stability of the latitudinal trend of the equinoctial asymmetry of the SqH. Moreover, there is obvious fluctuation in coefficient B, implying the variability of the reversal latitude under different levels of the equinoctial asymmetry of the SqH.

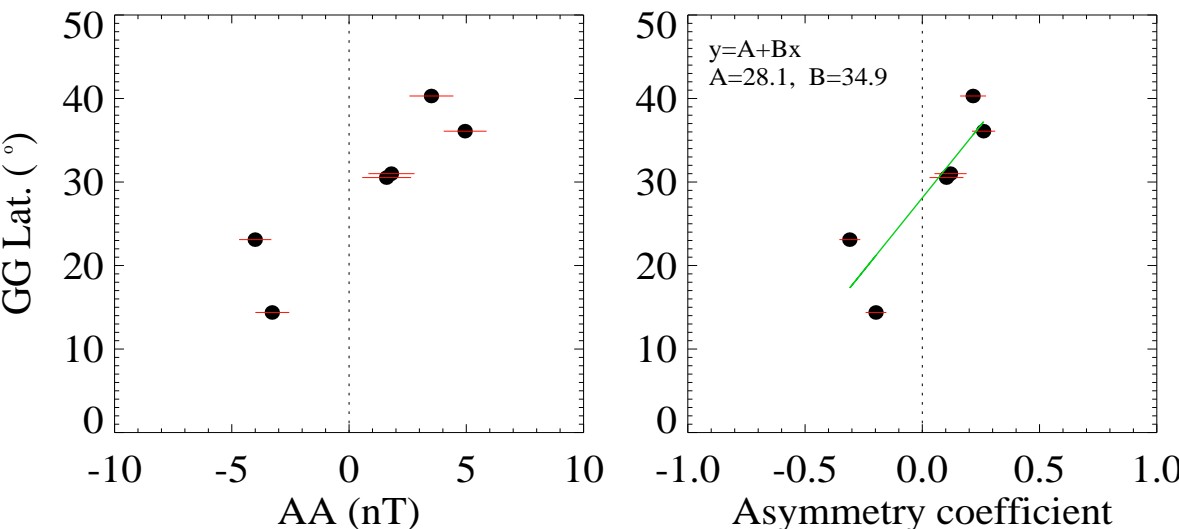

**Figure 7.** Dependence of the averaged equinoctial asymmetry AA and averaged asymmetry coefficient on geographic latitude. Red bars represent the SEM. Green curve in the right panel is the linear fit on the asymmetry coefficient, with fitting parameters indicated in the corner.

## 4. Discussion

Wulf (1963, 1965) [27,28] reported opposite signs of the SqH equinoctial asymmetry at HON and TUC and suggested that the meridional shift of the Sq current focus could cause this asymmetry. For the month-to-month changes of the Sq current focus position, Gupta (1973) [17] examined the meridional movement of Sq foci during the solar maximum of 1958 and found month-to-month shifts with latitude for both the northern and southern Sq foci. Tarpley (1973) [18] detected that the northern focus shifts southwards during autumn months, then northwards in winter. He suggested that this seasonal movement might be caused by the semiannual variation of the electrojet amplitude, induced by different seasonal changes at equatorial and middle–high latitudes. Vichare et al. (2017) [43] also found a drift toward lower latitudes of the Sq focus from March to September.

However, in terms of ionospheric dynamic processes, equinoctial asymmetry of Sq variations shows a strong relationship with the geomagnetic field, ionospheric conductivity, and tidal wind fields [31,44]. The correlation of the asymmetry with seasonal variations of lower-thermospheric winds and lower-ionospheric conductivity was discussed in numerous studies [27–30]. Obvious equinoctial asymmetry of electron and ion temperatures, ionospheric plasma density, and plasma drifts were also reported [45–49] and might be related to the marked equinoctial asymmetry of thermospheric winds [47,50]. Additionally, the asymmetry of vertical plasma drifts—mainly driven by E and F regions' neutral wind dynamo processes—has been observed and simulated [51–53]. Ren et al. (2012) [53] suggested that equinoctial asymmetry of the migrating semidiurnal tide and semiannual oscillation of the migrating diurnal tide in the tropical mesosphere–lower thermosphere region might be the principal drivers of the vertical plasma drift asymmetry.

In this study, we have shown that the SqH equinoctial asymmetry does not depend on solar activity (Figure 6). This implies a possible connection with regional factors caused by the ionospheric dynamo process. As is known, the Sq variation and the equivalent ionospheric current both show a strong longitudinal dependence [54–56]. Simulating the combined effects of the geomagnetic axis tilt and the Sq current magnetic anomaly, Xu et al. [55] suggested that longitudinal dependence of the Sq current is mainly caused by the geomagnetic field. This result was confirmed by Sager et al. [57]. Moreover, Ren et al. [53] analyzed the effect of the geomagnetic field on longitudinal variations of ionospheric vertical plasma drifts. They simulated ionospheric vertical plasma drifts driven by the migrating diurnal tide at 600 km around the equinoxes. Their results showed

a significantly weaker longitudinal asymmetry in a coaxial central dipole field than in a realistic geomagnetic field, suggesting the importance of the geomagnetic field in the ionospheric dynamo process.

In summary, the SqH equinoctial asymmetry is a complex phenomenon with multiple causes, such as the latitudinal movement of the Sq current focus, lower thermospheric winds, and the ionospheric dynamo process. In this study, we found that the sign of the equinoctial asymmetry is generally stable at a given observatory—positive north of the Sq current focus and negative at southern observatories.

## 5. Conclusions

In this study, measurements of the H component of the geomagnetic field acquired at six magnetometer observatories along the 120° E meridian chain in the years 1957–2013 were used to investigate the Sq range equinoctial asymmetry. Monthly values of the SqH range dA show large year-to-year variations at each observatory, but yearly averaged Sq range values (*R*) are comparable at all observatories. Equinoctial asymmetry is generally significant and positively correlated with geographic latitude. We identify three main features. First, the sign of the equinoctial asymmetry is opposite on the northern and southern sides of the Sq current focus, but the asymmetry is consistent at all observatories on the same side of the focus. SqH range maxima occur in spring at southern observatories but in autumn at northern observatories. Second, equinoctial asymmetry varies linearly with geographic latitude, increasing with increasing distances from the Sq current center, and its sign reverses at around 28.1° N, located near the focus of the average Sq current. This reversion latitude fluctuates in a range of 18 to 38 degrees when different confidence intervals of asymmetry coefficients are considered in the fitting procedure, reflecting the variability of the reversal latitude under different levels of the equinoctial asymmetry of the SqH. Third, the range and sign of the SqH equinoctial asymmetry do not appear to depend on solar activity. This suggests a possible origin in regional factors related to the ionospheric dynamo process. Further research is needed to investigate such issues as the latitudinal properties of the asymmetry at other longitudes or mechanisms that determine the sign of the asymmetry.

**Author Contributions:** Conceptualization, Y.W. and L.L.; methodology, Y.W.; software, Y.W.; validation, Y.W., L.L. and Z.R.; formal analysis, Z.R.; investigation, Y.W.; resources, Y.W.; data curation, Y.W.; writing—original draft preparation, Y.W.; writing—review and editing, L.L. and Z.R.; visualization, Y.W.; supervision, L.L.; project administration, Y.W.; funding acquisition, Z.R. All authors have read and agreed to the published version of the manuscript.

**Funding:** This research was funded by the National Key R&D Program of China (Grant No. 2018YFC1503504-03), IEF Grant of CEA (Grant No. 2021IEF0709), and the APSCO Earthquake Research Project Phase II and ISSI-BJ project: Integrating Satellite and Ground Observations for Earthquake Signatures and Precursors, the National Natural Science Foundation of China (41774161), the Open Research Project of Large Research Infrastructures of CAS—"Study on the interaction between low/mid-latitude atmosphere and ionosphere based on the Chinese Meridian Project", and the Meridian Project of China provided funding in the geomagnetic chain, respectively.

**Institutional Review Board Statement:** Not applicable.

**Informed Consent Statement:** Not applicable.

**Data Availability Statement:** The data used in this study are available from the World Data Centre for Geomagnetism in Kyoto, Edinburgh, and GFZ: http://wdc.kugi.kyoto-u.ac.jp, ftp.nmh.ac.uk, and ftp://ftp.gfz-potsdam.de/pub/home/obs/Kp_ap_Ap_SN_F107 (accessed on 18 September 2021).

**Acknowledgments:** The authors are grateful for all available data from the respective organizations. We thank Wenyao Xu of IGGCAS for his comments and suggestions, and we are sincerely grateful to the reviewers for their time, effort, and constructive remarks regarding the improvement of our work.

**Conflicts of Interest:** The authors declare no conflict of interest.

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
