# Peer review of "Equinoctial Asymmetry in Solar Quiet Fields along the 120° E Meridian Chain"

_applsci, doi:10.3390/app11199150_

Round 1
Reviewer 1 Report
The paper is well written and advance in the understanding of the equinoctial asymmetry, it should be published.
Author Response
We would like to thank you for your reviewing this paper and valuable comments/suggestions, which greatly advance the presentation of our manuscript.

Reviewer 2 Report
In their manuscript titled “Equinoctial asymmetry of the SqH field along the 120oE Meridian Chain”, Wu et al. present a quantitative analysis of the seasonal variability in the geomagnetic Sq field over five solar cycles (1957 – 2013). Based on measurements of the geomagnetic field’s horizontal component from six magnetic observatories located along the 120oE meridian chain, the study aims to draw the latitudinal profile of the equinoctial asymmetry and explore possible fluctuations related to the level of solar activity.
The presented results provide evidence pointing towards the sign of the equinoctial asymmetry being opposite northern and southern of the Sq current system centre in the northern hemisphere. In addition, the asymmetry was found to become more pronounced close to the Sq current system centre.
I am wondering, however, whether this study could benefit from exploring the latitudinal distribution of the geomagnetic field’s east-west component, D (usually referred to as the declination). In the past, El Hawary et al. (2012) have also explored the annual and semiannual variations in Sq of both the H and D component along the 96o MM MAGDAS chain of ground magnetometers.
Ιn general, the manuscript is well organised and written, presenting a wide introduction to the seasonal variability of geomagnetic Sq fields, a good description of techniques used on a long time series of ground magnetometer data. The results are interesting providing new insights on the complex dynamics of the Earth's magnetosphere-ionosphere system and a different viewpoint on some previous results.
I have no particular objection to the study as this has been carried out and believe that this work deserves to be published in Applied Sciences subject to the specific point indicated above.
There are some issues with the English language use and several typographical errors. For example:
In the Abstract:
“It shows larger values around spring than autumn at southern observatories, …” would more correctly read: “It shows larger values around spring than autumn equinox at southern observatories. And later, “and the converse is seen at northern observatories“, because “the converse” the noun means the opposite, but “converse” the verb means to have a conversation with someone.
It is not clear to me, and perhaps the reader, what is meant by the following sentence: “… Sq current focus, suggesting the stronger sensitive of the distant observatories than observatories close to the centre.”
In the Introduction:
Wu et al, list pair of terms that are used interchangeably but they are actually different. It is perhaps preferable to adopt throughout the manuscript one of the two terms proposed. For example, it would be more correct to use the term “range” than “amplitude” because only periodic fluctuations with a bounded range have an amplitude and this is half the difference between the minimum and maximum value.
It is customary for acronyms to be expanded at first mention in a manuscript. With this in mind, the acronym SqH, which is short for solar quiet (Sq) variations in the horizontal (H) component, should be moved to the Abstract and the title could read ”Equinoctial asymmetry in solar quiet fields along the 120oE Meridian chain.
In Data and Calculations:
It is my understanding that the Muntinlupa station is part of the 210 Magnetic Meridian (MM) chain. It is included in the catalogue of stations found here: https://stdb2.isee.nagoya-u.ac.jp/mm210/station.html. The acronym GZH should be short of Guangzhou instead of Zaoqing, both of which are part of the Chinese Meridian Network. Could you please specify the network to which stations in Table 1 belong and provide adequate references for each project?
In Table 1, it should read time coverage instead of dates, shouldn’t it? And in Figure 2, shouldn’t it read “Top: Yearly coverage of the observations …”?
Later on, it would more correctly read “Figure 2 shows the variation of solar radio flux at 10.7 cm (f10.7) in solar flux units (1 sfu = 10-22Wm-2Hz-1 … ”
In Figure 3 and within the text, would it be possible to describe how the left-hand side panels have been produced? Have variations of SqH been superposed per months for all years? What is the resolution in LT and time of year? Would it be possible to add grid lines at dawn, noon and dusk as well as at the solstices and equinoxes?
In Figure 4, I agreed with the authors that there is good agreement between the f10.7 index and both A and Aannual. Could a legend be added to indicate which colour line corresponds to which parameter of three A. Aannual and dA? The authors could also provide calculated correlation coefficient offering a quantitative measure of the association between three variables.
In Results:
In Figure 5, please replace the curly dash usually used to indicate that one number is approximately equal to another with an en dash (-) usually used to indicate a range of values.
Later on, it reads “…generally positive at northern observatories (with a few exceptional years) …. In what way where these years exceptional?
Later on, it reads “Both quantities increase with geographic latitude and with increasing distance between the observatory and the Sq current system centre, … “. Figure 7 does provide evidence pointing towards the equinoctial asymmetry being dependent on the distance from the Sq current system centre. However, the linear fit is relatively weak and does not clearly support the conclusion. Have the authors tried to plot the linear fit the distance from the Sq current system centre against the asymmetry coefficient?
References
El Hawary et al. (2012): Annual and semi-annual Sq Variations at 96o MM MAGDAS, Earth Planets Space, doi:10.5047/eps.2011.10.013
Reviewer 3 Report
The manuscript is of interest. However, several improvements are requested.
In conclusion, the manuscript is not suitable for publication in the present form but requires major revisions.

Round 2
Reviewer 3 Report
Please, see the attached revision#2.pdf file.

Round 3
Reviewer 3 Report
Review report
Review #3 of the manuscript: Equinoctial Asymmetry of the SqH Field along the 120°E Meridian Chain, by Yingyan Wu, Libo Liu and Zhipeng Ren, submitted to Applied Sciences.
General comments
The authors addressed most of the reviewer's requests. In particular, the reviewer would like to thank the authors for showing dA obtained by using different window sizes in the moving average procedure.
The manuscript is now suitable for publication after few minor revisions.
Minor comments
1) In the caption of Fig. 5 the SD should be shortly indicated.
2) Lines 193-195: Please, carefully check for typos and replace “sqrt” with the appropriate mathematical operator.
3) Line 230: “And it” in “And it..”
4) Line 341: Taking the results from new Fig. 7 the reversal latitude of 28.1° has a confidence bound of ~23-34° (or higher considering different fits) that the authors should include in their conclusions.
